# Clinical characteristics of re-positive COVID-19 patients in Huangshi, China: A retrospective cohort study

Ji Zhou[1,2☯], Jingying Zhang[1,3☯], Juan Zhou[1,4☯], Honggang Yi[5☯], Zichen Lin[6], Yu Liu[7], Min Zhu[6], Hongyu Wang[6], Wei Zhang[8], Hai Xu[8], Hangping Jiang[9], Zhengzhong Xiang[10], Ze Qu[11], Yuemei Yang[6], Linjuan Lu[6], Shuai Guo[6], Heng Fu[6], Ian M. Adcock[12], Yu Wei[3]*, Xin Yao[1,2]*

1 Department of Respiratory and Critical Care Medicine, The First Affiliated Hospital of Nanjing Medical University, Nanjing, China, 2 Huangshi Hospital of Traditional Chinese Medicine, Huangshi, China, 3 Department of Respiratory Medicine, Jiangsu Province Hospital of Chinese Medicine, Affiliated Hospital of Nanjing University of Chinese Medicine, Nanjing, China, 4 Department of Respiratory and Critical Care Medicine, Affiliated Hospital of Nantong University, Nantong, China, 5 Department of Biostatistics, School of Public Health, Nanjing Medical University, Nanjing, China, 6 Department of Internal Medicine, Nanjing Medical University, Nanjing, China, 7 Jurong Center for Disease Control and Prevention, Zhenjiang, China, 8 Department of Radiology, The First Affiliated Hospital of Nanjing Medical University, Nanjing, China, 9 Department of Cardiology, Huangshi Hospital of Youse, Huangshi, China, 10 Department of Respiratory, Huangshi Hospital of Youse, Huangshi, China, 11 Department of Medical Service, Huangshi Hospital of Youse, Huangshi, China, 12 Airway Disease Section, National Heart and Lung Institute, Imperial College London, London, United Kingdom

☯ These authors contributed equally to this work.
* yaoxin@njmu.edu.cn (XY); jsszyyhxk@163.com (YW)

**Data Availability Statement:** Data cannot be shared publicly because of identifying and sensitive patient information in this manuscript (including gender, age, patient contact history, medication

## Abstract

A cluster of patients with coronavirus disease 2019 (COVID-19) underwent repeated positive severe acute respiratory syndrome coronavirus 2 (SARS-CoV-2) RNA tests after they were discharged from the hospital. We referred to them as re-positive (RP) patients in this study. We aimed to describe the clinical characteristics of these patients in a retrospective cohort study. After being treated for COVID-19, the patients underwent 14 days of quarantine following their discharge from the Huangshi Hospital of Traditional Chinese Medicine and the Huangshi Hospital of Youse. Two additional sequential SARS-CoV-2 RNA tests were performed at the end of quarantine. The median age of the 368 patients was 51 years, and 184 (50%) patients were female. A total of 23 RP patients were observed at follow-up. Using multivariate Cox regression analysis, risk factors associated with RP included a higher ratio of lymphocyte/white blood cell on admission (adjusted HR 7.038; 95% CI, 1.911–25.932; P = 0.0034), lower peak temperature during hospitalization (adjusted HR, 0.203; 95% CI, 0.093–0.443; P<0.0001), and the presence of comorbidities, particularly hypertension or chronic diseases in the respiratory system (adjusted HR, 3.883; 95% CI, 1.468–10.273; P = 0.0063). Antivirus treatment with arbidol was associated with a lower likelihood of re-positive outcomes (adjusted HR, 0.178; 95% CI, 0.045–0.709; P = 0.0144).

history, incubation period and personal disease history). Data are available from the Ethics Committee of Huangshi Hospital of Traditional Chinese Medicine for researchers who meet the criteria for access to confidential data.The telephone number is 86-0714-6220894 and the E-mail address is 1250918038@qq.com.

**Funding:** The authors received no specific funding for this work.

**Competing interests:** The authors have declared that no competing interests exist.

## Introduction

Coronavirus disease 2019 (COVID-19) was first reported in December 2019 in Wuhan, China, and was declared a pandemic by the World Health Organization (WHO) on March 11, 2020. According to national reports received by the WHO, 972,303 cases of COVID-19 had been diagnosed worldwide by April 3[rd], and it affected more than 180 countries and caused 50,322 deaths [1]. The mortality of COVID-19 differed in studies (2.3%-4.3%) [2, 3], and it is less than that seen with the severe acute respiratory syndrome (SARS) and Middle East respiratory syndrome (MERS) [4]. However, the socioeconomic burden of SARS coronavirus 2 (SARS-CoV-2) far outweighs that of SARS and MERS. The basic reproduction number (R0) of SARS-CoV-2 was estimated to be between 2 and 3, suggesting that it is highly contagious [5]. Transmission from a carrier has been reported, and it is very important to screen and quarantine both infected patients and potential carriers [6].

Both SARS-CoV-2 and SARS-CoV belong to the Sarbecovirus subgenus, and their S glycoproteins have more than 80% amino acid sequence identity [7]. SARS-CoV-2 is a type of coronavirus derived from animals. It has been reported that SARS-CoV-2 has 86.9% genetic homology with bat SARS-like coronavirus. Beta-coronaviruses, including SARS-CoV-2 and MERS-coronavirus, can infect humans and result in severe pneumonia [8]. When we were treating patients with COVID-19 in Huangshi, we found that some patients had re-tested positive via nucleic acid detection after being discharged from the hospital. This is consistent with the results reported by Lan [9]. This phenomenon aroused our interest, and we called these patients "re-positive (RP) patients". It has also been reported that other viral infections cause recurrent positive nucleic acid tests after treatment including human papillomavirus (HPV) and hepatitis C virus (HCV) [10, 11]. This could make the disease chronic or cause large-scale transmission to a susceptible population.

We diagnosed and treated patients with COVID-19 according to the Chinese management guidelines for COVID-19 (version 2.0–6.0) as the guidelines updated [12]. Patients with an absence of fever for at least 3 days, improvement in chest CT, clinical remission of respiratory symptoms, and two sequential oropharyngeal swab samples tested negative for SARS-CoV-2 RNA obtained at least 24 hours apart met the criteria for discharge. Every discharged patient needed to remain quarantined at the designated place for 14 days. All patients would receive oropharyngeal swab again at the end of the quarantine period.

In this study, a total of 368 patients of the Huangshi Hospital of Traditional Chinese Medicine (TCM) and Huangshi Hospital of Youse, 84.4% of 436 COVID-19 discharged patients in the Huangshi city zone before March 1[st], were followed up to demonstrate the characteristics of RP patients.

## Methods

### Study design and participants

This was a retrospective cohort study of 368 patients aged 16 to 89 years with confirmed COVID-19 hospitalized at Huangshi Hospital of Traditional Chinese Medicine and Huangshi Hospital of Youse. All patients were diagnosed with COVID-19 according to the Chinese management guidelines for COVID-19 (version2.0–6.0). We followed up all discharge patients in these two hospitals between January 23th (the first admitted patient) and March 1[st]. The study was approved by the Research Ethics Commission of Huangshi Hospital of Traditional Chinese Medicine and Huangshi Hospital of Youse. The requirement for informed consent from study participants was waived by the Ethics Commission.

## Data collection

We extracted epidemiological, demographic, clinical, laboratory, and treatment data from electronic medical records using a standardized data collection form (S1 File). All data were independently checked by 3 physicians (HW, MZ, and ZL). A researcher (JZ) evaluated any differences in interpretation between the 3 primary reviewers.

## Laboratory procedures

**1. RT-PCR of SARS-CoV-2.** The oropharyngeal-swabs of patients were collected by well-trained medical staff. The samples were subsequently tested using quantitative reverse-transcription PCR to detect SARS-CoV-2 RNA. The open reading frame 1ab (ORF1ab) and nucleocapsid protein (N) were the two target genes. Total nucleic acid was extracted within 2 h using the respiratory sample RNA isolation kit (Shanghai BioGerm Medical Biotechnology Co Ltd) according to the manufacturer's instructions. The reaction mixture contained 5 μL of RNA template, 12 μL of reaction buffer, and 4 μL of reverse transcriptase mixture. Four microliters of probe primer solution were prepared for target gene amplification and tested using a 2019-nCoV nucleic acid detection kit according to the manufacturer's protocol (Shanghai Bio-Germ Medical Biotechnology Co Ltd). The positive and negative control groups were set. PCR conditions consisted of reverse transcription at 50˚C for 10 min, pre-denaturation at 95˚C for 5 min, followed by 40 cycles of denaturation at 95˚C for 10 s, annealing, extension, and collecting fluorescence signal at 55˚C for 40 seconds. The RT-PCR cycle threshold values were collected. The Ct value correlates with the number of copies of the virus in an inversely proportional. A Ct value of less than 38 was defined as a positive test result. These diagnostic criteria were based on the recommendations of the National Institute for Viral Disease Control and Prevention (China). The sequences for the ORF1ab real-time RT-PCR were as follows: forward primer: `CCCTGTGGGTTTTACACTTAA`; reverse primer: `ACGATTGTGCATCAGCTGA`; probe: `5'-FAM-CCGTCTGCGGTATGTGGAAAGGTTATGG-BHQ1-3'`. The sequences for the N gene were as follows: forward primer: `GGGGAACTTCTCCTGCTAGAAT`; reverse primer: `CAGACATTTTGCTCTCAAGCTG`; probe:5'-FAM-`TTGCTGCTGCTTGACAGATT-TAMRA-3`.

**2. CT Scan.** Every included patient received at least 2 chest CT scans: the first one at admission and the second one before discharge. Each RP patient underwent a further chest CT scan to evaluate any radiological abnormalities at re-admission. More CT scans were performed for some subjects if the physician thought it was necessary. Chest imaging results were reviewed separately by two radiologists (HX, a senior thoracic radiologist with 10 years' experience, and WZ, a senior thoracic radiologist with 10 years' experience) to assess image progression or absorption. Two evaluators independently assessed the chest CT features of the patient without access to clinical or laboratory findings. After separate evaluations, any disagreements were resolved by discussion and consensus.

**3. Routine blood tests.** Routine blood examinations included complete blood count, coagulation profile, serum biochemical tests, myocardial enzymes, and inflammation biomarkers. The results included white blood cell count(WBC), lymphocyte count(LY), neutrophil count(NE), platelet count(PLT), aspartate transaminase (AST), alanine transaminase (ALT), lactate dehydrogenase (LDH), C-reactive protein (CRP), erythrocyte sedimentation rate (ESR), B-type brain natriuretic peptide (BNP), and activated partial thromboplastin time (APTT).

## Definitions

Patients were considered to have a fever if their axillary temperature was raised to at least 37.3˚C. Exposure history was based on records of exposure to people with confirmed SARS--CoV-2 infection or had visited Wuhan. The severity of COVID-19 infection was defined

according to the Chinese management guidelines for COVID-19 (version 6.0). Chronic respiratory diseases in our study included previously diagnosed chronic obstructive pulmonary disease, asthma, and bronchiectasis. Chronic diseases in our study included hypertension, coronary heart disease, cancer, chronic renal disease, liver disease, hyperthyroidism, hypothyroidism, anemia, hyperlipidemia, arthrolithiasis and chronic respiratory diseases.

## Clinical management

Supplemental oxygen would be given to those with oxygen saturation dropped below 93% or patients felt obvious chest tightness. Patients clinically suspected of having community-acquired pneumonia were administered empirical broad-spectrum antibiotics and/or oral oseltamivir. Different anti-SARS-CoV-2 therapies, such as arbidol, lopinavir/ritonavir, ribavirin, ganciclovir, chloroquine, and α-interferon (IFN) nebulization were prescribed to selected patients at the physicians' discretion. As the role of glucocorticoids in COVID-19 treatment is controversial, their use was restricted to most patients given prednisone at a dose of 1 mg per kilogram of body weight for 1 week or less.

## Statistical analysis

Descriptive analyses of the variables were represented by median (IQR, 1st and 3rd), or counts and percentages (%). The rate with 95% confidence intervals (CIs) of the recurrence of positive test after the first discharge for COVID-19 patients was based on the binomial distribution. Differences in the distributions of laboratory indices between admission and discharge of the same patients were reported using differences with 95% CIs and the P values of the Wilcoxon signed-rank test. Univariate and multivariable Cox proportional hazard ratio (HR) models were used to assess HRs and 95% CIs as well as the P values with Wald tests, between individual factors on the recurrence of positive SARS-Cov-2 RNA test in patients with COVID-19. Proportional assumptions for the Cox proportional hazard model were examined using scaled Schoenfeld residuals. A stepwise selection method was used to select independent risk factors that affect outcomes.

The sample size varied because of missing data. The analyses regarding different factors were based on non-missing data, and missing data were not imputed. All tests were two-sided, and a P value less than 0.05 was considered statistically significant. All statistical analyses were performed using R software, version 3.5.1 (R Foundation for Statistical Computing).

## Results

### Demographic and clinical characteristics of all included patients

Before the end of the follow-up period on March 15th, we screened all 413 patients discharged before March 1st in Huangshi Hospital of Traditional Chinese Medicine and Huangshi Hospital of Youse. Of the 413 patients, 22 died during hospitalization, 17 were transferred between two hospitals and 6 were transferred to other hospitals. These 22 dead patients and 6 transferred patients were excluded from the study. The second admission of 17 patients transferred between the two hospitals was also excluded in the analysis. Cough and chest tightness worsened in two discharged patients during the quarantine. They were re-tested for SARS-CoV-2 immediately and showed positive results. At the end of the quarantine, the remaining 366 discharged patients showed no obvious symptoms. 21 of them had positive SARS-CoV-2 RNA tests at the end of quarantine. A total of 23 RP patients were included in this study as a re-positive group and the remaining 345 patients were included as a non-RP control group (flow chart can be seen in S1 Fig).

The demographic and clinical characteristics of all included patients are shown in Table 1. The median age in our cohort was 51 years, and the number of men and women was equal (184 vs. 184). Most of these patients (96.17%) were mild to moderate cases and 14 (3.83%) were severe and critical cases. About 91 (24.73%) of them had visited Wuhan, and 107 (29.16%) had contacted Wuhan residents. Of the patients, 12.50% declared contact with confirmed COVID-19 patients. The median incubation period in our study was 5 days. Common symptoms were fever (82.07%), cough (80.71%), chest tightness (45.11%), and 46.59% had fever on admission. During hospitalization, 74.66% of patients developed fever, and most of them were not above 39.0˚C. About 39.40% of the patients had comorbidities. Except for chronic respiratory diseases (n = 78, 21.20%), the most common comorbidities were hypertension (n = 68, 18.48%), diabetes (n = 32, 8.70%), and coronary heart disease (n = 15, 4.08%). The number of patients with one chronic disease was 84 (22.83%) and the number of patients with two or more chronic diseases was 32 (8.70%). Most patients received anti-viral therapy (arbidol, lopinavir/ritonavir, ribavirin, and ganciclovir).

## Laboratory indices in COVID-19 patients from admission to discharge

Laboratory markers were tracked from admission to discharge (Table 2). Physicians increased or decreased the laboratory tests undertaken according to each patient's condition, which resulted in fewer results for some patients. The number of subjects providing samples for each test is also listed in Table 2. When these patients were discharged from the hospital, their white blood cell count, lymphocyte count, and neutrophil and platelet counts were significantly elevated compared to the results upon admission. Biochemical indices, including AST and LDH, were significantly decreased at discharge. In addition, infection-related biomarkers such as CRP and ESR were also significantly decreased. There were no statistical differences in troponin I, B-type brain natriuretic peptide (BNP), and D-dimer levels between admission and discharge. APTT levels were significantly higher on admission and improved on discharge.

## Univariate analysis in patients with and without RP

The results of univariate analysis of the clinical characteristics between RP and non-RP patients are shown in Table 3. Decreased probability of fever during hospitalization(HR 0.22, 95% CI 0.09–0.51; P = 0.0005), lower temperature on admission (HR 0.51, 95% CI 0.28–0.93; P = 0.0291), and lower peak temperature (HR 0.24, 95% CI 0.11–0.49; P = 0.0001) were significantly associated with test-retest positivity. RP patients also showed a longer hospitalization time (HR 0.69, 95% CI 0.60–0.79; P <0.001).

Laboratory tests of RP and non-RP patients at admission were also compared (Table 4). The data showed a significant association of lymphocyte count (HR 2.13, 95% CI 1.05–4.30; P = 0.0353), lower levels of AST (HR0.94, 95% CI 0.90–0.99; P = 0.0286), LDH (HR 0.99, 95% CI 0.98–1.00; P = 0.0105), CRP (HR 0.96, 95% CI 0.93–1.00; P = 0.0390), ESR (HR 0.97, 95% CI 0.95–1.00; P = 0.0317) and APTT on admission (HR 0.92, 95% CI 0.86–0.97; P = 0.0042) in RP subjects. The level of D-Dimer was higher in RP group (HR 1.02, 95% CI 1.01–1.04; P = 0.0003). No significant difference was observed between the RP and non-RP groups at discharge (S1 Table).

## Multivariate analysis of patients with and without RP

Multivariate Cox regression models showed that several risk factors related to increased likelihood of RP included higher lymphocyte/white blood cells on admission (adjusted HR 7.038, 95% CI, 1.910–25.932; P = 0.0034), lower peak temperature during hospitalization (adjusted HR 0.203, 95% CI 0.093–0.443; P <0.0001), and the presence of comorbidities, particularly

**Table 1. The demographic and clinical characteristics of all included patients.**

| | Total |
|---|---|
| **No. of patients** | 368 |
| **Age, years** | 51.00(22.00, 40.00–62.00) |
| <65 | 296(80.43) |
| ≥65 | 72(19.57) |
| **Sex** | |
| Female | 184(50.00) |
| Male | 184(50.00) |
| **Exposure to source of transmission within past 14 days** | |
| Recently visited Wuhan | |
| No | 277(75.27) |
| Yes | 91(24.73) |
| Had contact with Wuhan residents | |
| No | 260(70.84) |
| Yes | 107(29.16) |
| Had contact with the confirmed COVID-19 patients | |
| No | 322(87.50) |
| Yes | 46(12.50) |
| Family gathering history | |
| No | 347(94.29) |
| Yes | 21(5.71) |
| **Incubation period, days** | 5.00(5.00, 3.00–8.00) |
| **Length of hospitalization, days** | 17.00(7.00, 14.00–21.00) |
| **Initial signs and symptoms** | |
| Fever | 302(82.07) |
| Cough | 297(80.71) |
| Sputum production | 121(32.88) |
| Chest tightness | 166(45.11) |
| Diarrhea | 40(10.87) |
| Headache | 62(16.85) |
| Nasal congestion | 14(3.80) |
| Chills | 30(8.15) |
| Sore throat | 48(13.04) |
| Myalgia or arthralgia | 28(7.61) |
| **Fever on admission** | 171(46.59) |
| median (IQR), ˚C | 37.10(1.00, 36.70–37.70) |
| <37.3˚C | 196(53.41) |
| 37.3–38.0˚C | 124(33.79) |
| 38.1–39.0˚C | 42(11.44) |
| >39.0˚C | 5(1.36) |
| **Fever during hospitalization** | |
| Yes | 274(74.66) |
| Peak temperature, median (IQR), ˚C | 37.90(1.30, 37.20–38.50) |
| <37.3 | 93(25.34) |
| 37.3–38.0 | 119(32.43) |
| 38.1–39.0 | 117(31.88) |
| >39.0 | 38(10.35) |
| **Severity** | |

(*Continued*)

**Table 1.** (Continued)

| | Total |
|---|---|
| Mild-Moderate | 352(96.17) |
| Severe | 11(3.01) |
| Critical | 3(0.82) |
| **Comorbidities** | |
| Any | 145(39.40) |
| Hypertension | 68(18.48) |
| Diabetes | 32(8.70) |
| Coronary heart disease | 15(4.08) |
| Cancer | 9(2.45) |
| Chronic renal disease | 8(2.17) |
| Liver disease | 14(3.80) |
| Chronic Respiratory diseases | 78(21.20) |
| Other diseases | 68(18.48) |
| Patients with one chronic disease | 84(22.83) |
| Patients with two or more chronic diseases | 32(8.70) |
| **Treatment** | |
| Anti-virus, Arbidol | 329(89.40) |
| Anti-virus, Lopinavir/Ritonavir | 51(13.86) |
| Anti-virus, Ribavirin | 86(23.37) |
| Anti-virus, Ganciclovir | 34(9.24) |
| Anti-virus, Arbidol + Lopinavir / Ritonavir | 48(13.04) |
| α-interferon nebulization | 215(58.42) |
| Inhaled corticosteroid | 36(9.78) |
| Systemic corticosteroid | 56(15.22) |

Data are described as number(%) or median (IQR, 1st and 3rd).

**Table 2. Changes of laboratory indices of patients during hospitalization.**

| | On Admission | | On Discharge | | Median of the Difference (95% CI) | P |
|---|---|---|---|---|---|---|
| | No. of patients tested | Value, median (IQR, 1st and 3rd) | No. of patients tested | Value, median (IQR, 1st and 3rd) | | |
| **WBC, ×10⁹ per L** | 368 | 4.48 (1.91, 3.55–5.46) | 345 | 5.40 (1.88, 4.45–6.33) | -0.86 (-1.11,-0.72) | $<$**0.0001** |
| **LY,×10⁹per L** | 368 | 1.17 (0.59, 0.89–1.48) | 345 | 1.55 (0.69, 1.18–1.87) | -0.31 (-0.36,-0.26) | $<$**0.0001** |
| **NE, ×10⁹per L** | 368 | 2.66 (1.66, 1.92–3.58) | 345 | 3.23 (1.53, 2.38–3.91) | -0.47 (-0.67,-0.31) | $<$**0.0001** |
| **PLT, ×10⁹per L** | 368 | 159.50 (70.00, 132.00–202.00) | 345 | 235.00 (100.00, 192.00–292.00) | -72.00 (-78.13,-59.57) | $<$**0.0001** |
| **AST, U/L** | 358 | 29.00 (15.00, 23.00–38.00) | 308 | 24.00 (15.00, 19.00–34.00) | 4.00 (2.46,9.27) | $<$**0.0001** |
| **ALT, U/L** | 356 | 22.00 (19.00, 16.00–35.00) | 307 | 31.00 (39.00, 19.00–58.00) | -6.00 (-15.52,-6.32) | $<$**0.0001** |
| **Troponin I, ng/ml** | 134 | 0.01 (0.01, 0.01–0.02) | 105 | 0.01 (0.01, 0.01–0.02) | 0.00 (-0.00,0.01) | 0.0993 |
| **BNP, pg/ml** | 71 | 20.40 (49.00, 10.00–59.00) | 61 | 22.00 (35.44, 10.00–45.44) | 0.00 (-3.99,70.60) | 0.0667 |
| **CRP, mg/L** | 343 | 12.31 (20.17, 6.18–26.35) | 291 | 1.64 (3.93 0.75–4.68) | 8.91 (12.83,19.25) | $<$**0.0001** |
| **ESR, mm/h** | 207 | 44.00 (40.00, 26.00–66.00) | 111 | 36.00 (52.00, 20.00–72.00) | 9.50 (2.84,14.08) | **0.0029** |
| **D-Dimer, μg/mL** | 233 | 0.19 (0.25, 0.05–0.30) | 176 | 0.20 (0.39, 0.11–0.50) | 0.00 (-1.36,3.93) | 0.2587 |
| **APTT, s** | 327 | 37.00 (8.30, 33.50–41.80) | 205 | 33.40 (5.50, 31.10–36.60) | 3.65 (3.00,4.78) | $<$**0.0001** |

P values were calculated using the Wilcoxon signed-rank test.

Table 3.  Univariate analysis of clinical characteristics among patients with and without RP.

| | Non-RP group | RP group | HR (95% CI) | P value |
|---|---|---|---|---|
| | no. (%)(*n* = 345) | no. (%)(*n* = 23) | | |
| **Age, median (IQR), years** | 50.00(23.00, 39.00–62.00) | 51.00(16.00, 42.00–58.00) | 1.00(0.97,1.03) | 0.7759 |
| <65 | 275(79.71) | 21(91.30) | ref. | |
| ≥65 | 70(20.29) | 2(8.70) | 0.40(0.09,1.69) | 0.2103 |
| **Sex** | | | | |
| Female | 168(48.70) | 16(69.57) | ref. | |
| Male | 177(51.30) | 7(30.43) | 0.53(0.22,1.30) | 0.1669 |
| **Exposure to source of transmission within past 14 days** | | | | |
| Recently visited Wuhan | | | | |
| No | 260(75.36) | 17(73.91) | ref. | |
| Yes | 85(24.64) | 6(26.09) | 0.75(0.29,1.92) | 0.5503 |
| Had contact with Wuhan residents | | | | |
| No | 244(70.72) | 16(72.73) | ref. | |
| Yes | 101(29.28) | 6(27.27) | 0.67(0.26,1.71) | 0.3993 |
| Had contact with the confirmed COVID19 patients | | | | |
| No | 304(88.12) | 18(78.26) | ref. | |
| Yes | 41(11.88) | 5(21.74) | 2.03(0.74,5.53) | 0.1682 |
| Family gathering history | | | | |
| No | 326(94.49) | 21(91.30) | ref. | |
| Yes | 19(5.51) | 2(8.70) | 1.60(0.37,6.92) | 0.5262 |
| **Incubation period, median (IQR), days** | 5.00(5.00, 3.00–8.00) | 6.00(13.00, 0.50–13.50) | 1.14(0.98,1.33) | 0.0933 |
| **Length of Hospitalization, median (IQR), days** | 17.00(7.00, 14.00–21.00) | 19.00(7.00, 15.00–22.00) | *0.69(0.60,0.79)* | *<0.0001* |
| **Initial signs and symptoms** | | | | |
| Fever | 282(81.74) | 20(86.96) | 1.25(0.37,4.22) | 0.7248 |
| Cough | 275(79.71) | 22(95.65) | 4.23(0.57,31.42) | 0.1592 |
| Sputum production | 114(33.04) | 7(30.43) | 0.76(0.31,1.85) | 0.5486 |
| Chest tightness | 153(44.35) | 13(56.52) | 1.35(0.59,3.12) | 0.4797 |
| Diarrhea | 37(10.72) | 3(13.04) | 0.70(0.21,2.38) | 0.5682 |
| Fatigue | 237(68.70) | 14(60.87) | 0.58(0.25,1.35) | 0.2079 |
| Headache | 57(16.52) | 5(21.74) | 1.15(0.43,3.10) | 0.7807 |
| Nasal congestion | 14(4.06) | 0.00(0.00) | . . . | . . . |
| Chills | 27(7.83) | 3(13.04) | 1.44(0.42,4.85) | 0.5613 |
| Sore throat | 45(13.04) | 3(13.04) | 0.73(0.22,2.48) | 0.6179 |
| Myalgia or arthralgia | 27(7.83) | 1(4.35) | 0.50(0.07,3.74) | 0.5023 |
| **Fever on admission** | | | | |
| Yes | 162(47.09) | 9(39.13) | 0.52(0.22,1.22) | 0.1339 |
| median (IQR), ˚C | 37.15(1.00, 36.70–37.70) | 36.90(1.00, 36.50–37.50) | *0.51(0.28,0.93)* | *0.0291* |
| <37·3˚C | 182(52.91) | 14(60.87) | ref. | |
| 37·3–38·0˚C | 115(33.43) | 9(39.13) | 0.78(0.33,1.82) | 0.5646 |
| 38·1–39·0˚C | 42(12.21) | 0.00(0.00) | . . . | . . . |
| >39·0˚C | 5(1.45) | 0.00(0.00) | . . . | . . . |
| **Fever during hospitalization** | | | | |
| Yes | 261(75.87) | 13(56.52) | *0.22(0.09,0.51)* | *0.0005* |
| Peak temperature, median (IQR), ˚C | 38.00(1.30, 37.30–38.60) | 37.40(1.10, 36.90–38.00) | *0.24(0.11,0.49)* | *0.0001* |
| <37·3 | 83(24.13) | 10(43.48) | ref. | |
| 37·3–38·0 | 110(31.98) | 9(39.13) | *0.40(0.16,1.00)* | *0.049* |
| 38·1–39·0 | 113(32.85) | 4(17.39) | *0.14(0.04,0.47)* | *0.0013* |

(*Continued*)

**Table 3.** (Continued)

| | Non-RP group no. (%)(n = 345) | RP group no. (%)(n = 23) | HR (95% CI) | P value |
|---|---|---|---|---|
| >39·0 | 38(11.05) | 0.00(0.00) | . . . | . . . |
| **Severity** | | | | |
| Mild-Moderate | 330(96.21) | 22(95.65) | ref. | |
| Sever | 11(3.21) | 0.00(0.00) | . . . | . . . |
| Critical | 2(0.58) | 1(4.35) | 3.04(0.40,23.16) | 0.2834 |
| **Comorbidities** | | | | |
| Any | 134(38.84) | 11(47.83) | 1.41(0.62,3.19) | 0.4124 |
| Hypertension | 63(18.26) | 5(21.74) | 1.31(0.48,3.53) | 0.5961 |
| Diabetes | 30(8.70) | 2(8.70) | 1.07(0.25,4.55) | 0.9319 |
| Coronary heart disease | 14(4.06) | 1(4.35) | 0.93(0.12,6.92) | 0.9425 |
| Cerebrovascular disease | 4(1.16) | 1(4.35) | 2.97(0.40,22.21) | 0.2882 |
| Cancer | 9(2·61) | 0(0·00) | . . . | . . . |
| Chronic renal disease | 7(2.03) | 1(4.35) | . . . | . . . |
| Liver disease | 12(3.48) | 2(8.70) | 2.85(0.38,21.31) | 0.3068 |
| Chronic Respiratory diseases | 74(21.45) | 4(17.39) | 1.81(0.42,7.77) | 0.4247 |
| Other diseases | 63(18.26) | 5(21.74) | 0.68(0.23,2.00) | 0.4839 |
| **Treatment in hospital** | | | | |
| Arbidol | 309(89.57) | 20(86.96) | 0.36(0.10,1.27) | 0.1129 |
| Lopinavir /Ritonavir | 47(13.62) | 4(17.39) | 0.76(0.26,2.26) | 0.6219 |
| Ribavirin | 77(22.32) | 9(39.13) | 1.72(0.74,3.99) | 0.2056 |
| Ganciclovir | 29(8.41) | 5(21.74) | 1.96(0.72,5.36) | 0.1903 |
| Arbidol + Lopinavir/Ritonavir | 45(13.04) | 3(13.04) | 0.55(0.16,1.88) | 0.3420 |
| α-interferon nebulization | 203(58.84) | 12(52.17) | 0.56(0.24,1.29) | 0.1721 |
| Inhaled corticosteroid | 35(10.14) | 1(4.35) | 0.62(0.21,1.84) | 0.3852 |
| Systemic corticosteroid | 52(15.07) | 4(17.39) | 1.26(0.17,9.37) | 0.8213 |

Data are described as number(%) or median (IQR, 1st and 3rd). P values were calculated using the Wilcoxon signed-rank test.

**Table 4. Univariate analysis of laboratory indices on admission of RP and non-RP group.**

| | Non-RP group(n = 345) | RP group(n = 23) | HR (95% CI) | P |
|---|---|---|---|---|
| **WBC, ×10⁹ per L** | **4.44(1.93, 3.51–5.44)** | **5.25(1.68 4.42–6.10)** | **1.18(0.93,1.50)** | **0.1666** |
| **LY, ×10⁹per L** | 1.16(0.58, 0.88–1.46) | 1.37(0.55, 1.20–1.75) | *2.13(1.05,4.30)* | *0.0353* |
| **Lymphocyte / White blood cell** | 0.27(0.14, 0.20–0.34) | 0.27(0.09, 0.24–0.33) | 4.87(0.09,274.96) | 0.4415 |
| **NE, ×10⁹per L** | 2.65(1.67, 1.90–3.57) | 3.03(1.81, 2.41–4.22) | 1.10(0.83,1.44) | 0.5067 |
| **PLT, ×10⁹per L** | 159.00(69.00, 132.00–201.00) | 173.00 (74.00, 140.00–214.00) | 1.00(1.00,1.01) | 0.1753 |
| **AST, U/L** | 29.00(15.00, 23.00–38.00) | 23.00(10.00, 20.00–30.00) | *0.94(0.90,0.99)* | *0.0286* |
| **ALT, U/L** | 23.00(20.00, 16.00–36.00) | 14.50(10.00, 13.00–23.00) | 0.96(0.93,1.00) | 0.0769 |
| **LDH, U/L** | 235.00(98.50, 193.50–292.00) | 190.00 (46.00, 182.00–228.00) | *0.99(0.98,1.00)* | *0.0105* |
| **Troponin I, ng/ml** | 0.01(0.01, 0.01–0.02) | 0.01(0.02, 0.01–0.03) | . . . | . . . |
| **BNP, pg/ml** | 20.40(55.00, 10.00–65.00) | 24.65(31.81, 1.09–32.90) | 0.99(0.96,1.01) | 0.3597 |
| **CRP, mg/L** | 12.43(20.68, 6.43–27.11) | 8.56(15.60, 5.22–20.82) | *0.96(0.93,1.00)* | *0.0390* |
| **ESR, mm/h** | 44.00(39.00, 27.00–66.00) | 32.50(32.00, 13.00–55.00) | *0.97(0.95,1.00)* | *0.0317* |
| **D-Dimer, μg/mL** | 0.19(0.25, 0.05–0.30) | 0.20(0.25, 0.10–0.35) | *1.02(1.01,1.04)* | *0.0003* |
| **APTT, s** | 37.20(8.20, 33.70–41.90) | 35.50(6.20 31.40–37.60) | *0.92(0.87,0.97)* | *0.0042* |

Data are median (IQR, 1st and 3rd). P values were calculated using Wald tests.

**Table 5. Multivariate analysis of patients with and without RP.**

| | HR (95% CI) | P value |
|---|---|---|
| Age, median (IQR), year(≥65 *vs* <65) | 0.395(0.083,1.875) | 0.2423 |
| Sex(Male *vs* Female) | 0.721(0.278,1.868) | 0.5004 |
| Comorbidities with Hypertension or Chronic respiratory disease (Yes *vs* no) | 3.883(1.468,10.273) | *0.0063* |
| Peak temperature during hospitalization(˚C) | 0.203(0.093,0.443) | *<0.0001* |
| Lymphocyte/White blood cell on admission | 7.038(1·910,25.932) | *0.0034* |
| Anti-virus, Arbidol (Yes *vs* no) | 0.178(0.045,0.709) | *0.0144* |

The Cox proportional hazard model was used to screen the important variables by stepwise regression based on re-positivity.

hypertension or chronic diseases in the respiratory system (adjusted HR 3.883, 95% CI 1.468–10.273; P = 0.0063). This analysis showed that arbidol reduced the probability of re-positive outcomes (adjusted HR 0.178, 95% CI 0.045–0.709; P = 0.0144). The results of multivariate analysis are listed in Table 5.

## Clinical characteristics of RP patient re-admission

Considering the potential infectious risks of RP patients, all 23 RP patients were re-admitted to the hospital for observation and treatment. Their clinical characteristics are shown in Table 6. None of the RP patients who were hospitalized for the second time had fever. Three patients

**Table 6. Clinical characteristics of RP patients in re-admission.**

| | No. (%) |
|---|---|
| No. of patients | 23 |
| Age, median (IQR), year | 51.00(16.00, 42.00–58.00) |
| <65 | 21(91.30) |
| ≥65 | 2(8.70) |
| Sex | |
| Female | 16(69.57) |
| Male | 7(30.43) |
| Initial signs and symptoms | |
| Fever | 0(0) |
| Cough | 3(13.04) |
| Sputum production | 0(0) |
| Chest tightness | 3(13.04) |
| Diarrhea | 0(0) |
| Headache | 0(0) |
| Nasal congestion | 0(0) |
| Chills | 0(0) |
| Sore throat | 1(4.55) |
| Myalgia or arthralgia | 0(0) |
| Evaluation of chest CT on re-admission | |
| No change | 3(13.04) |
| Partially absorbed | 13(56.52) |
| Completely absorbed | 7(30.43) |

Data are median (IQR, 1st and 3rd) or n (%).

(13.04%) complained of cough and chest tightness. Another patient (4.55%) reported a slight sore throat on the second admission.

Two radiologists independently assessed the chest CT scans of the re-positive patients. The images taken at re-admission were compared to those taken at the first discharge. No patient showed aggravated lung images: no obvious changes were seen in the lungs of 3 (13.04%) patients, 13 (56.52%) were partially absorbed, and 7 (30.43%) were almost completely absorbed.

RP patients showed no obvious abnormality in the laboratory measures, including routine blood tests, biochemical and inflammatory markers, and indicators of coagulation function (S2 Table). Fifteen RP patients tested for antibodies to SARS-CoV-2, all of them showed positive antibody responses (S3 Table).

## Discussion

This retrospective cohort study identified the clinical features of RP COVID-19 patients in Huangshi, China. These patients had a lower peak temperature and presented with higher lymphocyte counts but lower levels of AST, LDH, CRP, and APTT, and more comorbidities of hypertension or chronic diseases in the respiratory system compared with patients without RP. There was no difference in demographics, epidemiological features, and treatment between the RP and non-RP groups by univariate analysis.

Since the RP patients were well quarantined in our study before re-admission and all RP patients who received antibody tests showed positive results, we considered that the incidence of re-infection with SARS-CoV-2 was low [13]. Therefore, we suspect that there are other reasons for the recurrent positive RNA tests. To the best of our knowledge, an accurate explanation for RP is lacking. In Singapore, the duration of SARS-CoV-2 viral shedding from nasopharyngeal aspirates was prolonged up to 24 days (median duration 12 days; range 1–24 days) after symptom onset and towards the end of this period, the virus was only intermittently detected [14]. Zhou and colleagues reported that detectable SARS-CoV-2 RNA persisted for a median of 20 days in survivors and that it was sustained until death [15]. There are several possible mechanisms such as intermittent viral shedding, false negatives during the initial discharge, residual viral presence, and viral distribution to explain the existence of RP patients [14, 16, 17]. We suggest that our RP patients may have a longer SARS-CoV-2 viral shedding time and/or intermittent viral shedding. However, we also noticed that one study from Hong Kong, researchers could not amplify active viruses from samples of re-positive patients [18]. Thus, it is also possible that the presence of re-positive patients was caused by broken virus fragments or dead viruses that were not completely cleared in the body.

At the same time, some researchers believe that the low efficiency of nasopharyngeal swab detection might result in the appearance of test-retest positivity [19]. According to the current study, the viral load of SARS-CoV-2 gradually decreases in the rehabilitation stage [20]. In our opinion, the occurrence of some RP patients is indeed partly related to this factor. Two of the RP patients in our study who were positive for SARS-CoV-2 RNA in the middle of 14 day-quarantine were more likely due to false-negative results before discharge. However, we believe that not all RP patients can be explained in this way. Kang et al. reported that 3.3% of the 8922 patients were re-positive in Korea, and these patients represented only minor symptoms [21]. According to the results of our and other published studies on RP, patients with RP have unique clinical characteristics.

RP patients showed lower peak temperatures and lower levels of CRP, ESR, and APTT. These findings suggest that RP patients might be in a weak state of the immune response. It is widely known that high fever and increased CRP levels are signs of an activated immune status. Interestingly, Wu and colleagues found that high fever was positively associated with the

development of acute respiratory distress syndrome (ARDS) but negatively related to death
[22]. An effective innate immune response is important in fighting viral infections, which
relies heavily on interferon (IFN) type I responses. For SARS-CoV and MERS-CoV, the type I
IFN response to viral infection is suppressed [23]. The RP patients in our study had lower peak
temperatures, lower CRP levels, and reduced presence of fever, which indicates that they
might have a reduced anti-viral immune response. Severe cases of COVID-19 tend to have
lower lymphocyte counts and higher plasma levels of LDH and TNF-$\alpha$ [24, 25]. Biopsies of
COVID-19 infected lung also showed a heightened inflammatory response in patients who
died [26]. Furthermore, a cytokine storm has been reported in patients with SARS who die
with significantly elevated levels of IL-18, IP-10, MIG, and MCP-1 [27]. It is postulated, there-
fore, that an overactive immune response drives disease progression.

In contrast, in children, who possess an immature immune system, the infection rate of
SARS-CoV-2 is lower and the symptoms are milder compared to adults [28]. The recurrent
positive group in our study with lower maximum temperature, lower incidence of fever, and
lower LDH levels, indicating that these RP patients have a weaker response to SARS-CoV-2
akin to that seen in children.

In our study, multivariant analysis in our study showed that arbidol might reduce the recur-
rent SAR-CoV-2 positive rate. Indeed, arbidol has a direct antiviral effect during the early viral
replication of SARS-CoV in vitro [29]. In addition, one retrospective cohort study has shown
that combined arbidol and Lopinavir/ritonavir treatment shortens the duration of viral shed-
ding compared to lopinavir/ritonavir alone [30].

We analyzed RP patients upon re-admission and found no obvious abnormality in their
blood tests. There were no physical complaints, and chest CT showed no signs of deterioration
compared with the previous scans. These data indicated that although the nucleic acid test was
again positive, RP patients were clinically similar to those in continuous recovery.

Our study has some limitations. First, at the outset of the outbreak of COVID-19 in Hubei
province, medical resources were lacking, and we could not repeat the RNA test every day to
determine the dynamic changes in viral titer. We could not perform the antibody test in every
patient. Second, measurement of serum levels of cytokines such as IL-2, IFN-,$\alpha$ and TNF-$\alpha$
that might help us understand the immune response in patients were not examined. Further
studies are required to detail the infectivity and immune status of these patients.

In conclusion, the exact causes of recurrent positive test results for SARS-CoV-2 are still
unknown. Due to its highly contagious nature, it is better for us to be cautious when discharg-
ing patients with the potential to shed the virus. To our knowledge, this is the first cohort
study to describe the characteristics of RP patients with COVID-19. To prevent potential fur-
ther transmission by these RP patients, it is reasonable to quarantine patients for at least 14
days after discharge and to re-test them again before contact with other people is allowed. Fur-
ther research on RP patients will help us better understand the process of infection, clearance,
and metabolism of SARS-CoV-2 in patients. It also helps us to better deal with the spread of
COVID-19.

## Supporting information

**S1 Table. Univariate analysis of laboratory indices of patients with and without RP at dis-
charge.**
(DOC)

**S2 Table. Laboratory indices of RP patients at re-admission.**
(DOC)

**S3 Table. Antibody results of 15 RP-patients.**
(DOC)

**S1 Fig. Flow chart of the overview of this study.**
(TIF)

**S1 File.**
(DOC)

**S2 File.**
(DOCX)

## Author Contributions

**Conceptualization:** Ian M. Adcock, Yu Wei, Xin Yao.

**Data curation:** Ji Zhou, Jingying Zhang, Honggang Yi, Zichen Lin, Yu Liu, Min Zhu, Hongyu Wang, Wei Zhang, Hai Xu, Hangping Jiang, Zhengzhong Xiang, Ze Qu, Yuemei Yang, Linjuan Lu, Shuai Guo, Heng Fu.

**Formal analysis:** Ji Zhou, Jingying Zhang, Honggang Yi, Yu Liu.

**Investigation:** Wei Zhang, Hai Xu, Zhengzhong Xiang, Ze Qu, Yuemei Yang, Linjuan Lu, Yu Wei.

**Methodology:** Honggang Yi, Yu Liu, Wei Zhang, Hai Xu, Yu Wei, Xin Yao.

**Software:** Honggang Yi, Yu Liu, Hai Xu.

**Supervision:** Honggang Yi, Yu Wei, Xin Yao.

**Validation:** Jingying Zhang, Hai Xu, Yu Wei.

**Writing – original draft:** Ji Zhou, Jingying Zhang, Juan Zhou.

**Writing – review & editing:** Jingying Zhang, Ian M. Adcock, Xin Yao.

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
