## [Decision Letter · Decision Letter 0]

29 Jul 2020

PONE-D-20-18154

Clinical characteristics of Re-positive recovering patients with COVID-19 in Huangshi, China: a retrospective cohort study

PLOS ONE

Dear Dr. Yao,

Thank you for submitting your manuscript to PLOS ONE. After careful consideration, we feel that it has merit but does not fully meet PLOS ONE’s publication criteria as it currently stands. Therefore, we invite you to submit a revised version of the manuscript that addresses the points raised during the review process.

We look forward to receiving your revised manuscript.

Kind regards,

Wenbin Tan

Academic Editor

PLOS ONE

Journal Requirements:

2.We suggest you thoroughly copyedit your manuscript for language usage, spelling, and grammar. If you do not know anyone who can help you do this, you may wish to consider employing a professional scientific editing service.  

3. Thank you for stating in the text of your manuscript "the requirement for informed consent from study participants was waived by the Ethics Commission". Please also add this information to your ethics statement in the online submission form.

4. Please revise your tables to replace p-values of 0.0000 with p-values of <0.0001.

5.We note that you have indicated that data from this study are available upon request. PLOS only allows data to be available upon request if there are legal or ethical restrictions on sharing data publicly. For information on unacceptable data access restrictions, please see http://journals.plos.org/plosone/s/data-availability#loc-unacceptable-data-access-restrictions.

6. Please include your tables as part of your main manuscript and remove the individual files. Please note that supplementary tables (should remain/ be uploaded) as separate "supporting information" files

Additional Editor Comments:

This manuscript reported some cases of re-positive COVID-19 patients after discharge. I have some comments as follows.

(1) The authors need to discuss whether re-positive patients after recovery was due to second infection or they were still contiguous? An antibody result will help to address this question.

(2) The term of "re-positive recovering (RPR) patients" is awkward; do you mean re-positive patients after recovery or discharge?

(3) Formats of all numbers in abstract, contents and tables need to change, for example, changing "P=0·0015" into "P=0.0015"

(4) in many tables, it was stated that median (IQR) was used; however, i am doubtful for that. Did you mean S.D. since there was only one digit presented? For IQRs, 1st and 3rd IQR are generally used. Please change all the values into median (IQR, 1st and 3rd) in all of the tables.

(5) in the discussion, authors stated that "... they might have a reduced anti-viral immune response." I disagree on it. These patients most likely were due to residual viral loads, second likely re-infection.

Reviewers' comments:

Reviewer's Responses to Questions

**Comments to the Author**

1. Is the manuscript technically sound, and do the data support the conclusions?

Reviewer #1: Yes

Reviewer #2: Yes

2. Has the statistical analysis been performed appropriately and rigorously? 

Reviewer #1: Yes

Reviewer #2: Yes

3. Have the authors made all data underlying the findings in their manuscript fully available?

Reviewer #1: Yes

Reviewer #2: Yes

4. Is the manuscript presented in an intelligible fashion and written in standard English?

Reviewer #1: Yes

Reviewer #2: No

5. Review Comments to the Author

Reviewer #1: The authors report a retrospective cohort study to investigate the re-positive recovering patients in COVID 19 patients. They reported RPR included a younger age, higher level of lymphocytes on admission, shorter fever duration and more comorbidities of hypertension and chronic diseases in respiratory system. They claim RPR patients had a weaker and shorter inflammatory response.

This is a well thought of and properly presented study on a relevant clinical question.

However, on closer inspection, I have a few concerns:

1. A research from South Korea reported 292 re-positive COVID 19 cases, they determined the re-positive is due to the limits of test method, and the remnants of the virus was picked up in the test. What do you think about it?

2. Another research in Wuhan claim that the positive rate of virus with pharyngeal swabs in COVID 19 patients is only 32% (126/398). Viral load changes as the disease progresses, false negative result at discharge may affect the ratio of RPR in COVID 19 patients.

Reviewer #2: Zhou et al presented a study on COVID-19 RPR patients by using various laboratory techniques and statistical analysis. The manuscript is well written, but few typos and grammar issues exist. I have the following comments:

1. The introduction is very brief, it would be nice if authors can add more about SARS-CoV-2 in it.

2. Why only 383 patients were chosen for the study?

3. Line 86- If the patients were tested positive during the discharge period, why they kept in quarantine instead of continuing with the same treatment? Please explain.

4. What were the parameters taken while data collection as authors mentioned a data collection form, it would be beneficial for readers if authors can provide the sample form in the supplementary.

5. Among the comorbidities, most of the focus is on hypertension and chronic diseases in respiratory system; while as it would have been interesting to see RPR in the immunocompromised such as HIV positive patients. Authors have completely ignored this factor, however it will be an interesting inclusion.

6. PLOS authors have the option to publish the peer review history of their article (what does this mean?). If published, this will include your full peer review and any attached files.

Reviewer #1: No

Reviewer #2: **Yes: **Aijaz Ahmad

---

## [Author Response · Author response to Decision Letter 0]

14 Sep 2020

Dear editor: 

Thanks a lot for having reviewed our manuscript.   Now we have revised the manuscript according to the reviewers’ comments. Most of the revisions are in the manuscript. Some explanations regarding the revisions of our manuscript are as follows. During the revison, we found that a few patients were transfered between two hospitals and excluded them in new manuscript. Thank you very much for your comments. Your questions have played a very important role in improving this research.

Our manuscript was re-edited by Editage (www.editage.com) on your recommendation.

Answers to Editor:

(1)The authors need to discuss whether re-positive patients after recovery was due to second infection or they were still contiguous? An antibody result will help to address this question.

Reply：Thank you very much for your suggestion. We reviewed data and looked up the relevant information about the antibody results. Because our patients were followed-up earlier (Before March 1), only 15 re-positive patients in all 23 re-positive patients were tested for antibodies. All these 15 patients showed positive antibody results (2 patients with IgM positive, 15 patients with IgG positive; detailed information can be found in S3 Table). There is one article named “Humoral Immune Response to SARS-CoV-2 in Iceland” published in the New England Journal recently. Their results indicate that antiviral antibodies against SARS-CoV-2 did not decline within 4 months after diagnosis [1]. In rhesus macaques，researchers found out a relationship between a humoral immune response to SARS-CoV-2 infection and protection against reinfection by this virus [2]. To our knowledge, there was no article reported re-infection of SARS-Cov-2 in such a short time in the presence of positive antibody. Therefore, we speculate that the re-positive patients in this study are less likely to be caused by the second infection. According to your advice, we have added the antibody results and the discussion of re-infection in the manuscript. 

(2) The term of "re-positive recovering (RPR) patients" is awkward; do you mean re-positive patients after recovery or discharge?

Reply：In our original vision, we wanted to describe the clinical characteristics of asymptomatic re-positive patient , so we defined such a description. However, we had to admit that this description is really awkward. After repeated discussions in our group， we replaced the term “re-positive recovering (RPR) patients” with “re-positive (RP) patients ” in new manuscript and two symptomatic re-positive patients were also included in study .

(3) Formats of all numbers in abstract, contents and tables need to change, for example, changing "P=0·0015" into "P=0.0015"

Reply：We have made improvements in accordance with your requirements.

(4)in many tables, it was stated that median (IQR) was used; however, i am doubtful for that. Did you mean S.D. since there was only one digit presented? For IQRs, 1st and 3rd IQR are generally used. Please change all the values into median (IQR, 1st and 3rd) in all of the tables.

Reply：Thanks for your advice. We have changed our values in tables as you mentioned.

(5) in the discussion, authors stated that "... they might have a reduced anti-viral immune response." I disagree on it. These patients most likely were due to residual viral loads, second likely re-infection.

Reply：Thank you for your question. One study from Hong Kong, researchers could not amplify active viruses from samples of re-positive patients [3]. Thus, it is possible that the presence of re-positive patients was likely to be caused by broken virus fragments or dead viruses that are not completely cleared in the body. And talking about re-infection, we considered the possibility was relatively low. Firstly，15 of 23 re-positive patients underwent antibody test, and all of them showed positive results. There was no article reported re-infection of SARS-Cov-2 in such a short time in the presence of positive antibody. Secondly, all patients were very strictly quarantined after discharge from hospital within 14 days. They had no chance to exposure to the source of the infection. So, we considered that the possibility of re-infection was relatively low. However, we think your opinion is very reasonable, and we had re-writen the discussion part, which put the first reason of re-positive results as residual viral loads.

Answers to Review 1

1.A research from South Korea reported 292 re-positive COVID 19 cases, they determined the re-positive is due to the limits of test method, and the remnants of the virus was picked up in the test. What do you think about it?

Reply：Thank you very much for this question. Your two comments on this article are very important. We had been thinking about the two questions you raised.

We had read the article you mentioned. It is a brief report based on the data given by South Korea Centers for Disease Control and Prevention. Most of the information mentioned in the article is consistent with the results of our article, such as most of the re-positive patients showed mild symptoms and were concentrated in relatively young age groups. 

In fact, we were also very interested in what causes the re-positive results in recovered patients. Regrettably, it was strictly limited in management of covid-19 patients’ specimens here. We could not conduct specific studies on that by ourselves and can only learn from other researchers’ article. According to the current results，we believed that the author’s description of dead virus or virus fragments in re-positive patients was a highly likely. However, to confirm what caused this phenomenon in re-positive patients requires further research. We will continue to monitor related article. 

2.Another research in Wuhan claim that the positive rate of virus with pharyngeal swabs in COVID 19 patients is only 32% (126/398). Viral load changes as the disease progresses, false negative result at discharge may affect the ratio of RPR in COVID 19 patients.

Reply：The question you raised is very important.

In the progression of covid-19 patients’ infection, their viral load had experienced a rapid increase in the early stage of the disease to a slow decrease in the recovery period. Pharyngeal swabs were not the best choice from the perspective of detection accuracy. But pharyngeal swabs had the advantages of convenient collection, it was the main method for specimen collection in the early stage. In order to maximize the reliability of results, we required that the nucleic acid collectors in each ward be a fixed number of experienced medical staff in Huangshi's covid-19 management. At the same time, we required that each patient underwent two independent nucleic acid tests before being discharged from the hospital, with an interval of more than 24 hours. Through these two methods, we tried our best to improve the accuracy of the nucleic acid results in the research. And thanks to your advice, we re-discussed the issue in our manuscript to make readers be aware of the possibility of false negative result.

Answers to Review 2:

1.The introduction is very brief, it would be nice if authors can add more about SARS-CoV-2 in it.

Reply：Thank you for your suggestion. We had added introduction of SARS-CoV-2 in article in revised manuscript.

2.Why only 383 patients were chosen for the study?

Reply：The number of patients we included in the study was not actually set by us. After we determined the deadline for follow-up, we screened all patients from the two participating units before the deadline. All eligible patients have been included in the study. In terms of proportion, 84.4% of the patients discharged from hospital before March 1st in Huangshi City have been included in this study, which is a good representative.

3. Line 86- If the patients were tested positive during the discharge period, why they kept in quarantine instead of continuing with the same treatment? Please explain.

Reply：We are sorry that our description in the article is not clear enough. All discharged patients will be quarantined for 14 days. After the quarantine period, everyone would undergo nucleic acid testing. Negative patients could go home, and positive patients will be admitted to the hospital to continue treatment. In the revised manuscript, we described this process more meticulously.

4. What were the parameters taken while data collection as authors mentioned a data collection form, it would be beneficial for readers if authors can provide the sample form in the supplementary.

Reply：We really appreciate your suggestion. In the revised manuscript, we provided sample form in supplementary file.

5.Among the comorbidities, most of the focus is on hypertension and chronic diseases in respiratory system; while as it would have been interesting to see RPR in the immunocompromised such as HIV positive patients. Authors have completely ignored this factor; however, it will be an interesting inclusion.

Reply：Thank you very much for the question. It is really important. We are also very interested in the re-positive performance of patients with abnormal immune status. In our study, all the patients when they admitted to hospital were inquired of detailed personal history, including HIV infection. Due to the limited conditions, not all the patients were screened for HIV test during hospitalization. According to your advice, we screened the data again and we found that none of the patients had a history of AIDS. Among all the patients, 63 of them underwent HIV antibody test due to possible blood transfusion, and all of their results were negative. Then we searched the literatures. The epidemiological study from Huangshi Center for Disease Control and Prevention showed that there were 1 518 HIV/AIDS cases reported from 2011 to 2017 in Huangshi city [4], while the total population of Huangshi is 2.69 million. The infection rate of HIV in Huangshi is relatively low. That could be the reason why HIV patients were rare in our study. 

References:

1. Gudbjartsson DF, Norddahl GL, Melsted P, Gunnarsdottir K, Holm H, Eythorsson E, et al. Humoral Immune Response to SARS-CoV-2 in Iceland. N Engl J Med. United States; 2020; 

2. Deng W, Bao L, Liu J, Xiao C, Liu J, Xue J, et al. Primary exposure to SARS-CoV-2 protects against reinfection in rhesus macaques. Science. 2020;369:818–23. 

3. Kang H, Wang Y, Tong Z, Liu X. Retest positive for SARS-CoV-2 RNA of “recovered” patients with COVID-19: Persistence, sampling issues, or re-infection? J Med Virol. 2020; 

4. Fu Xiong, Ruiqing Xie, Xianzhou Ke ZQ. AIDS epidemic and epidemic characteristics in Huangshi, 2011- 2017. China Trop Med [Internet]. China Tropical Medicine; 2018;18:906. Available from: http://www.cntropmed.com/CN/abstract/article_13504.shtml

---

## [Decision Letter · Decision Letter 1]

13 Oct 2020

PONE-D-20-18154R1

Clinical characteristics of re-positive COVID-19 patients in Huangshi, China: a retrospective cohort study

PLOS ONE

Dear Dr. Yao,

Thank you for submitting your manuscript to PLOS ONE. After careful consideration, we feel that it has merit but does not fully meet PLOS ONE’s publication criteria as it currently stands. Therefore, we invite you to submit a revised version of the manuscript that addresses the points raised during the review process.

We look forward to receiving your revised manuscript.

Kind regards,

Wenbin Tan

Academic Editor

PLOS ONE

As the reviewer commented below, the ethical statements addressing the patients cohorts from both institutions are very important. It must be clarified.

Reviewer #3: 1. In this study. 383 patients in Huangshi hospital of Traditional Chinese Medicine (TCM) and Huangshi hospital of Youse was included, but the Ethics Statement showed that the study was approved only by the Research Ethics Commission of Huangshi Hospital of Traditional Chinese Medicine. It should has another Ethics Statement from Huangshi hospital of Youse. Please provide the Ethics statement to address this issue. 

2. In this study, all patients were diagnosed with COVID-19 according to the Chinese management guideline for COVID-19 (version 6.0). This guideline was published in the Feb. 18th 2020, how could all patients be diagnosed according to this guideline? Some patients must be diagnosed with covid-19 before the guideline being published.

3. Two patients, whose cough and chest tightness worsened during quarantine after they were discharged, were considered as non-recovering patients who were not included in our study. But why they were considered as non-recovering patients, not the re-positive recovering (RPR) patients? They were negative when they were discharged from the hospital.

4. About 30·68% of them had comorbidities. The most common comorbidities were hypertension (n=72，18·8%) followed by diabetes (n=33,8·62%) and coronary heart disease(n=16, 4·18%). How many patients had chronic disease? And how many patients who had two comorbidities at least For example patients had hypertension and diabetes together.

5. Multivariate analysis showed that the arbidol reduced the probability of repositive outcome. But it did`t give the criteria for the use of drugs. Who had received the arbidol and who hadn't?

6. In Multivariate analysis, there showed “a younger age (adjusted HR 0·075, 95% CI 0·009-0·657; P =0·0193),” “the arbidol reduced the probability of repositive outcome (adjusted HR 0·048, 95% CI:0·010-0·231; P =0·0002).”, but in DISCUSSION, “There was no difference in demographics, epidemiological features and treatment between the RPR and non-RPR groups by univariate analysis.” Please clarify.

---

## [Author Response · Author response to Decision Letter 1]

20 Oct 2020

Dear editor: 

Thanks a lot for having reviewed our manuscript.   Now we have revised the manuscript according to the reviewers’ comments. Most of the revisions are in the manuscript. Some explanations regarding the revisions of our manuscript are as follows. Just as we communicated in the E-mail on Oct 15, comments of 3 and 6 from the reviewer # 3 were ignored here. 

Answers to Review #3

1.In this study. 383 patients in Huangshi hospital of Traditional Chinese Medicine (TCM) and Huangshi hospital of Youse was included, but the Ethics Statement showed that the study was approved only by the Research Ethics Commission of Huangshi Hospital of Traditional Chinese Medicine. It should has another Ethics Statement from Huangshi hospital of Youse. Please provide the Ethics statement to address this issue. 

Reply：Thank you very much for reminding. In fact, this study was approved by the ethics committees of two hospitals at the beginning of the study. We have modified the inaccurate description in the manuscript and provided the ethics committee approval documents issued by these two hospitals (form Line 103 to Line 104).

2.In this study, all patients were diagnosed with COVID-19 according to the Chinese management guideline for COVID-19 (version 6.0). This guideline was published in the Feb. 18th 2020, how could all patients be diagnosed according to this guideline? Some patients must be diagnosed with covid-19 before the guideline being published.

Reply：Our description here is not rigorous. We had made corrections in the manuscript (On Line 83 and Line 101).

3. Two patients, whose cough and chest tightness worsened during quarantine after they were discharged, were considered as non-recovering patients who were not included in our study. But why they were considered as non-recovering patients, not the re-positive recovering (RPR) patients? They were negative when they were discharged from the hospital.

Reply：This issue had been corrected in revised manuscript. Just as we communicated in the E-mail on Oct 15, this comment was ignored here. 

4. About 30·68% of them had comorbidities. The most common comorbidities were hypertension (n=72，18·8%) followed by diabetes (n=33,8·62%) and coronary heart disease(n=16, 4·18%). How many patients had chronic disease? And how many patients who had two comorbidities at least For example patients had hypertension and diabetes together.

Reply：Your comments are very valuable. We had added a description about this in Table 1and manuscript (From Line 170 to Line 172, Line 234 to Line 235).

5. Multivariate analysis showed that the arbidol reduced the probability of repositive outcome. But it did`t give the criteria for the use of drugs. Who had received the arbidol and who hadn't?

Reply：This question is very good. However, at the beginning of the year, in response to the unprepared attack of COVID-19, we did not have an accurate understanding of the therapeutic value of Abidol, or whether it was more effective in certain groups of people. You could find that our guideline has been updated from the 2st edition to the 6th edition within a few months, and our understanding of covid-19 has been improving and changing until now. Therefore, in this article, we have not stipulated what kind of patients should or should not use Arbidol. Each doctor in charge chose combination of drugs based on his or her professional knowledge.

6. In Multivariate analysis, there showed “a younger age (adjusted HR 0·075, 95% CI 0·009-0·657; P =0·0193),” “the arbidol reduced the probability of repositive outcome (adjusted HR 0·048, 95% CI:0·010-0·231; P =0·0002).”, but in DISCUSSION, “There was no difference in demographics, epidemiological features and treatment between the RPR and non-RPR groups by univariate analysis.” Please clarify.

Reply：This issue had been corrected in revised manuscript. Just as we communicated in the E-mail on Oct 15, this comment was ignored here.

---

## [Editor Report · Decision Letter 2]

23 Oct 2020

Clinical characteristics of re-positive COVID-19 patients in Huangshi, China: a retrospective cohort study

PONE-D-20-18154R2

Dear Dr. Yao,

We’re pleased to inform you that your manuscript has been judged scientifically suitable for publication and will be formally accepted for publication once it meets all outstanding technical requirements.

Kind regards,

Wenbin Tan, Ph.D.

Academic Editor

PLOS ONE
---

## [Editor Report · Acceptance letter]

27 Oct 2020

PONE-D-20-18154R2 

Clinical characteristics of re-positive COVID-19 patients in Huangshi, China: a retrospective cohort study 

Dear Dr. Yao:

I'm pleased to inform you that your manuscript has been deemed suitable for publication in PLOS ONE. Congratulations! Your manuscript is now with our production department. 

Kind regards, 

on behalf of

Dr. Wenbin Tan 

Academic Editor

PLOS ONE